# Responsiveness of Daily Life Gait Quality Characteristics over One Year in Older Adults Who Experienced a Fall or Engaged in Balance Exercise

**DOI:** 10.3390/s23010101

**Published:** 2022-12-22

**Authors:** Sabine Schootemeijer, Roel H. A. Weijer, Marco J. M. Hoozemans, Kim Delbaere, Mirjam Pijnappels, Kimberley S. van Schooten

**Affiliations:** 1Department of Human Movement Sciences, Faculty of Behavioural and Movement Sciences, Amsterdam Movement Sciences, Vrije Universiteit Amsterdam, Van der Boechorststraat 7, 1081 BT Amsterdam, The Netherlands; 2Department of Neurology, Center of Expertise for Parkinson & Movement Disorders, Donders Institute for Brain, Cognition and Behavior, Radboud University Medical Center, 6500 HB Nijmegen, The Netherlands; 3Department of Neurology, Leiden University Medical Center, 2300 RC Leiden, The Netherlands; 4Falls, Balance and Injury Research Centre, Neuroscience Research Australia, Sydney 2031, Australia; 5School of Population Health, University of New South Wales, Sydney 2052, Australia

**Keywords:** accidental falls, accelerometry, activity monitoring, aged, exercise, locomotion, mobility, wearable devices

## Abstract

Gait quality characteristics obtained from daily-life accelerometry are clinically relevant for fall risk in older adults but it is unknown whether these characteristics are responsive to changes in gait quality. We aimed to test whether accelerometry-based daily-life gait quality characteristics are reliable and responsive to changes over one year in older adults who experienced a fall or an exercise intervention. One-week trunk acceleration data were collected from 522 participants (65–97 years), at baseline and after one year. We calculated median values of walking speed, regularity (sample entropy), stability (logarithmic rate of divergence per stride), and a gait quality composite score, across all 10-s gait epochs derived from one-week gait episodes. Intraclass correlation coefficients (ICC) and limits of agreement (LOA) were determined for 198 participants who did not fall nor participated in an exercise intervention during follow-up. For responsiveness to change, we determined the number of participants who fell (*n* = 209) or participated in an exercise intervention (*n* = 115) that showed a change beyond the LOA. ICCs for agreement between baseline and follow-up exceeded 0.70 for all gait quality characteristics except for vertical gait stability (ICC = 0.69, 95% CI [0.62, 0.75]) and walking speed (ICC = 0.68, 95% CI [0.62, 0.74]). Only walking speed, vertical and mediolateral gait stability changed significantly in the exercisers over one year but effect sizes were below 0.2. The characteristic associated with most fallers beyond the LOA was mediolateral sample entropy (4.8% of fallers). For the exercisers, this was gait stability in three directions and the gait quality composite score (2.6% of exercisers). The gait quality characteristics obtained by median values over one week of trunk accelerometry were not responsive to presumed changes in gait quality after a fall or an exercise intervention in older people. This is likely due to large (within subjects) differences in gait behaviour that participants show in daily life.

## 1. Introduction

Despite strong evidence that falls can be prevented, the lack of valid and responsive fall risk instruments hampers the efficient allocation of interventions and accurate monitoring of their effectiveness. Moreover, supervised evaluations of fall risk are prone to several disadvantages, including the limited predictive ability of single fall risk measures [1], the costs of screening tools [1], the limited time of the healthcare personnel [2], and that the supervised evaluations do not take into account how someone behaves in daily life. Unsupervised monitoring with inertial measurement units may provide a relatively cheap and objective alternative that may be used to obtain representative evaluations of people’s gait quality in their daily lives. Previous studies showed that characteristics of daily life gait quality derived from inertial sensor data are reliable measures of fall risk [3] and are validated against fall incidence and time-to-fall [4,5,6,7,8]. They hold promise in identifying individuals at the highest risk for falls and detecting potential intervention effects [9]. It is unknown whether daily life gait quality characteristics are indeed responsive to clinically relevant changes.

Older people’s fall risk changes over time and gait quality may be responsive to these changes. People who have experienced (multiple) falls may experience injuries or develop a fear of falling, which impacts their daily activities and gradually deteriorates balance and gait [3,4,10]. On the other hand, exercise interventions reduce fall incidence rates and the number of older people experiencing a fall [11]. Several characteristics have been explored and a few consistently hold promise for fall-risk estimation. Walking speed is lower in people at high risk for falling [12]. Gait consistency, as indexed by sample entropy, is higher for vertical (VT) movements [13] and lower for mediolateral (ML) movements in people with higher fall risk [5]. Local instability, as indexed by the logarithmic rate of divergence per stride, is higher in people with higher fall risk [5]. Moreover, a gait quality composite score, consisting of characteristics of gait variability, intensity, and smoothness, has also been shown predictive of future falls [4,5]. There is evidence that gait characteristics obtained in laboratory settings are responsive to change: gait deteriorates over time in older adults [14] and can improve after intervention in older adults or people with Parkinson’s disease [15,16,17]. However, there is limited evidence of whether these effects are similar to daily life gait characteristics. Although the before-mentioned gait quality characteristics have been linked to fall risk, the responsiveness of these characteristics to negative events (i.e., falls) or positive interventions (i.e., physical training) still has to be established for scientific and clinical use. Therefore, the aim of this study was to establish whether gait quality characteristics derived from daily life trunk accelerometry are reliable and responsive to change because of a fall or an exercise intervention in older people during a one-year follow-up.

## 2. Materials and Methods

### 2.1. Participants

We analysed data of 522 participants from the Veilig in Beweging Blijven (VIBE) longitudinal cohort study [18] and the *StandingTall* randomised controlled trial [19,20]. Both studies included older adults who underwent assessment of daily life gait at baseline, one year later, and with a 12-month follow-up of falls. The *StandingTall* trial additionally provided half of its participants with a balance exercise intervention. The inclusion criteria for the VIBE cohort and *StandingTall* trial are provided in Appendix A Table A1. For this study, only individuals with complete gait data at baseline and after 12 months were included. Participants who experienced one or more falls and participated in the exercise intervention were excluded from our analysis. The protocol of the VIBE study was approved by the ethics committee of the faculty of Behavioural and Movement Sciences of the Vrije Universiteit Amsterdam (VCWE-2016-129). The protocol of the *StandingTall* randomised control trial was approved by the Human Research Ethics Committee, University of New South Wales, Sydney, Australia (HREC 14/266). All participants provided written informed consent.

We divided the 522 eligible participants into three groups (Figure 1): a reference group that did not fall and did not participate in the exercise intervention (*n* = 198), a fallers group that experienced one or more falls during follow-up but did not participate in the exercise intervention (*n* = 209), and an exercisers group that did not experience a fall but did participate in the exercise intervention (*n* = 115).

### 2.2. Participants’ Characteristics

Participants’ characteristics were assessed during a visit to the laboratory at the Vrije Universiteit, Amsterdam (The Netherlands) or Neuroscience Research Australia, Sydney (Australia). Fall history over the 12 months prior to the assessment was obtained during an interview or with a questionnaire. Participants’ global cognitive function was assessed with the Mini-Mental State Examination (MMSE) [21] in VIBE and the Montreal Cognitive Assessment (MoCA) [22] in the *StandingTall* intervention. The presence of depressive symptoms was assessed in VIBE with the Geriatric Depression Scale-15 (GDS-15; participants with a score equal to or greater than 6 were considered to have depressive symptoms) [23] and in *StandingTall* with the Patient Health Questionnaire-9 (PHQ-9; participants with a score equal to or greater than 10 were considered to have depressive symptoms) [24]. Isometric knee extension strength was measured with a unidirectional force transducer (KAP-E 2kN, A.ST. GmbH Dresden, Germany) in VIBE and as part of the Physiological Profile Assessment in *StandingTall* [25]. Handgrip strength was measured with a handgrip dynamometer (TKK 5401, Takei Scientific Instruments, Tokyo, Japan) in VIBE.

### 2.3. Falls

All participants completed a fall calendar and received monthly phone calls over a one-year follow-up period to determine whether they had fallen. A fall was defined as “an unexpected event in which the participant comes to rest on ground, floor, or lower level” [26]. Participants were classified as a faller when they experienced at least one fall during the follow-up.

### 2.4. Exercise Intervention

The intervention group in the *StandingTall* trial was asked to take part in a program with at least two hours per week of balance exercises. The *StandingTall* program is provided via a tablet application, can be performed in the comfort of one’s own home, is tailored to the individual’s balance ability, and increases in difficulty as the individual progresses over time [19,20].

### 2.5. Assessment of Daily Life Gait Quality

Participants wore a McRoberts MoveMonitor activity monitor (DynaPort MM+, McRoberts, The Hague, The Netherlands) for one week at baseline and after one year. Participants were instructed to wear the monitor on the lower back with the use of an elastic band for one week at all times, except for aquatic activities. The monitors collected trunk accelerations at a range of +/−6 g and were set to sample at a frequency of 100 Hz.

Gait quality characteristics were calculated from the trunk accelerometry data using custom programs in MATLAB R2018b (Mathworks, Natrick, MA, USA). The first six hours of data were excluded from our analysis to exclude data that could have been collected during transportation to and from our laboratories. Locomotion episodes of at least 10 s were identified from the acceleration signal using a classification algorithm validated by the manufacturer of the monitors [27]. These episodes were divided into epochs of 10 s and gait quality characteristics were calculated for each of these epochs.

Walking speed was estimated using step length and duration, utilizing a method described previously [28]. Sample entropy, a measure of gait regularity, was determined as described by Rispens and van Schooten [13]. Logarithmic rate of divergence per stride is a measure of gait stability, also referred to as local dynamic stability or Lyapunov exponent, which was calculated in vertical, mediolateral, and anteroposterior (AP) directions by dividing logarithmic rate of divergence per stride by stride duration [3]. We also extracted a gait quality composite score comprising four gait quality characteristics: root mean square of the acceleration signal in ML, index of harmonicity in ML, magnitude of the acceleration signal at dominant period in the frequency domain in AP, and autocorrelation of the acceleration signal at dominant period in the frequency domain in VT [5,9]. For every gait quality characteristic, we took the median of the distribution of outcomes over all 10-s epochs from one week, as a representation of the participant’s daily life gait quality.

As most gait quality characteristics are dependent on walking speed [5,29], we corrected the gait quality characteristics (GQ) for walking speed using the coefficients (offset α and slope β) of linear regression models between the individual gait characteristics as dependent variable and walking speed as independent variable [5,29].
GQcorrected = (GQ − α)/(β × walking speed)(1)

### 2.6. Statistics

Statistical analyses were performed in R [30]. Baseline descriptive characteristics were compared to determine if the three groups of interest were similar at baseline using linear regression, Kruskal–Wallis (for age), Chi-square (for sex and number of participants with a history of falls in the year prior to the studies), and Mann–Whitney U tests (for depressive symptoms). Post hoc pairwise comparisons were reported with Holm correction for multiple testing if there was a significant main effect. Fall risk, knee extension torque, cognitive function, depressive symptoms, and handgrip strength were assessed using different instruments and therefore reported and tested separately for the participants in the VIBE and *StandingTall* studies.

#### 2.6.1. Reliability of Gait Quality Characteristics

We determined the test-retest reliability and limits of agreement in the reference group, where we expected the gait quality characteristics (corrected for speed) to remain stable over time ([31], see Appendix B for formulas). Intraclass correlation coefficients (ICCs) exceeding 0.70 were considered acceptable [31].

#### 2.6.2. Responsiveness to Change

We performed paired samples *t*-tests in the reference group, the fallers, and the exercisers group to compare changes from baseline to follow-up for all gait quality characteristics. We also determined Cohen’s d effect sizes (*d* < 0.20 were considered a small effect, 0.2–0.5 moderate, and >0.80 large [31]). Given that the use of paired samples t-tests to determine responsiveness is limited [22], we also tested whether the change in gait quality characteristics in the fallers and exercisers was beyond the LOA using a one-sided independent samples t-test, assuming decline in the fallers and improvement in the exercisers. We further calculated what percentage of fallers and exercisers scored beyond the LOA to gain insight into how many people showed a change larger than expected based on time.

## 3. Results

### 3.1. Descriptive Characteristics

The descriptive characteristics of the reference-, fallers-, and exercise groups are shown in Table 1. At baseline, there were significant differences between the groups in age, body mass index (BMI), cognitive function, and fall history. Fallers were on average 1.2 (SD 7.9) years younger and exercisers 1.4 (SD 8.2) years older than the reference group (p_fallers_ = 0.03; p_exercisers_ =0.002; Table 1). Exercisers had a 1.4 (SD 6.7) kg/m^2^ higher BMI compared to the fallers (*p* = 0.03). Exercisers scored 0.8 (SD 3.1) points higher on the MoCA compared to the reference group (StandingTall only: *p* = 0.04). Fallers more often experienced a fall one year prior to baseline than the other groups (50.2% vs. 29.3% and 37.4%, *p* < 0.001).

### 3.2. Reliability of Gait Quality Characteristics

The ICCs for the agreement between baseline and follow-up were acceptable (exceeded 0.70) for all gait quality characteristics except for the logarithmic rate of divergence per stride in VT (ICC = 0.69, 95% confidence interval [0.62–0.75]) and for walking speed (ICC = 0.68, 95% confidence interval [0.62–0.74]) (Table 2). The LOA are presented in Table 2. One example of a Bland–Altman plot, for the gait quality composite score corrected for walking speed, is shown in Figure 2.

The percentage of fallers that showed a decrease in walking speed, sample entropy mediolateral direction (ML), or gait quality, or an increase in sample entropy in the vertical direction (VT), or logarithmic rate of divergence in VT, ML, or the anteroposterior direction are reported. For the exercisers, we reported the percentage of participants that changed beyond the opposite LOA.

### 3.3. Responsiveness

Walking speed decreased significantly in the exercise group although the effect size was small (t(114) = −3.81, *p* < 0.001, d = −0.22). Moreover, logarithmic rate of divergence per stride in VT (t(114) = −2.19, *p* = 0.03, d = −0.13) and in ML (t(114) = −2.02, *p* = 0.05, d = −0.10) significantly decreased after one year in the exercisers, indicating a small improvement in stability of the gait pattern (Table 3). No changes in any of the gait quality characteristics occurred in the other groups.

The change in gait quality characteristics over a year was not larger in the fallers and exercisers groups than the LOA we observed for the reference group over the same period (*p* > 0.05; Figure 3 and Appendix A Figure A1 and Figure A2).

Over a year, only a few fallers showed a deterioration of gait quality characteristics larger than the LOA, as indicated by a decrease in walking speed, sample entropy ML, or gait quality, or an increase in sample entropy in VT, or the logarithmic rate of divergence in VT, ML, or AP beyond the LOA of the reference group (Table 2). The percentage of fallers that changed beyond the LOA ranged between 1.91% and 4.78% and the characteristic with most fallers beyond the LOA was sample entropy ML. For the exercisers, we reported the percentage of participants that changed beyond the opposite LOA (Table 2). The percentage of exercisers that changed beyond the LOA was between 0.87% and 2.61%. Logarithmic rate of divergence in all three directions and gait quality were the characteristics with most participants beyond the LOA (all 2.61%).

## 4. Discussion

Our aim was to determine the reliability and responsiveness to a change in daily life accelerometry-based gait quality characteristics as a result of a fall or an exercise intervention in older people. We expected to find that a considerable proportion of fallers and exercisers would show changes in gait quality characteristics exceeding the LOA determined in the reference group over one year. The ICCs for agreement exceeded 0.70 for all gait quality characteristics except for the logarithmic rate of divergence per stride in VT and walking speed. Even though the logarithmic rate of divergence per stride in VT and ML changed significantly over a year, indicating an improvement in the exercisers, the effect sizes were small (*d* < 0.2). The LOA established in a reference group who did not experience a fall or participated in balance exercise was wide, and only a few fallers and exercisers showed a change beyond the LOA. These findings suggest that the gait quality characteristics are not responsive to the effects of experiencing a fall or participating in a balance exercise intervention.

The ICCs were acceptable (>0.7) for all gait characteristics except for the logarithmic rate of divergence in VT and walking speed, but were lower than those reported by our earlier study where we measured twice, two weeks apart [9]. Furthermore, the within-subject variability observed in this study between the two measurements in the control group with one year in-between was relatively large compared to the study where we measured two weeks apart [9]. The wide limits of agreement and small change over time in our reference group corroborates with Rojer and Coni [32], which showed that walking speed measured in daily life was robust over one year time but resulted in wide LOA. Nevertheless, intervention studies generally look at changes over longer periods such as a year, justifying our timeframe.

We expected a fall or an exercise intervention to significantly affect gait quality characteristics although this appeared not evident. Since we found few changes in gait quality characteristics over time with small effect sizes and few fallers or exercisers beyond the LOA, it remains inconclusive whether the gait quality characteristics were not responsive or whether the effect of experiencing a fall or participating in the exercise intervention was not large enough to be captured. Given that differences in gait quality characteristics between fallers and non-fallers and the effects of interventions are accompanied by effect sizes (Cohen’s d) between 0.3–5 [9], the latter might be the case. This could be explained by the fact that the *StandingTall* balance exercise improved standing balance after 12 months and reduced the falls and injurious falls rate over 24 months but did not significantly affect functional mobility, gait quality in the laboratory, or fall rates at 12 months [20]. The limited intervention effects at 12 months might not have been sufficient to result in changes in daily life gait quality. Future studies are required to assess the responsiveness to changes in daily gait quality characteristics due to exercise interventions that focus on balance during gait, or in clinical populations such as stroke survivors, where such interventions are expected to have greater effects.

Although this study is the first to investigate the responsiveness of daily life gait quality characteristics to changes after a fall or exercise intervention, some methodological considerations should be noted. First, the *StandingTall* intervention focused on balance and involved mainly standing balance and targeted stepping exercises. We expected improvements in balance to transfer to changes in gait quality but such transfers may be difficult [33]. Second, we included all people in the exercise group who did not fall, also those with low adherence to the intervention. Selecting only those with high adherence might have allowed us to observe larger effects. However, given that only a few people scored beyond the LOA we do not expect this adjustment to have led to a different conclusion. Third, we defined a faller as someone who experienced “at least one unexpected event in which the participant comes to rest on ground, floor, or lower level” [26]. Forty percent of the participants in our sample experienced at least one such event but recovery from one fall, especially if that fall did not result in trauma, may not necessarily lead to long-lasting (measurable) gait (quality) deficits. A fall could be due to a deterioration in gait quality that already occurred prior to the fall, effectively not introducing a change in gait quality. The effect of multiple or injurious falls on gait is, therefore, an area for further investigation. Fourth, we relied on self-reported falls, which could have led to recall bias. However, as participants in the VIBE cohort received monthly phone calls and participants in the *StandingTall trial* completed weekly online fall diaries and received a call if data were missing, we do not expect to have missed many falls. Fifth, we extracted gait quality as the median of all locomotion epochs over one week. We found a limited change in the median value of gait characteristics within a week from baseline to one-year follow-up but there may have been a change in the distribution of epochs of gait characteristics within the measurement weeks. For instance, the range of a gait characteristic per epoch within a week could have changed. Such changes would potentially not affect the median values, yet they could contain valuable information about a person’s gait behaviour [13]. Rispens and van Schooten in [13] showed that using the tenth and ninetieth percentile, representing gait in extreme situations, did not improve fall prediction compared to using the median. Future studies should consider the distribution of gait quality characteristics and not only focus on one (median) value over all walking epochs during a whole week to fully capture gait in real life. Lastly, we corrected the gait quality characteristics for walking speed since they are affected by walking speed [5,6,29], which we consider a strength of our study. Falling or participating in an exercise intervention could lead to changes in walking speed. Older adults who are concerned about falling might walk slower [34], while exercise and balance interventions might improve self-selected walking speed in the lab [35,36,37]. However, the *StandingTall* balance exercise intervention did not affect self-selected walking speed on a 10 m walk test in the lab [20]. Our results showed that walking speed in daily life did not change due to a fall and decreased slightly (and with only a Cohen’s *d* of −0.22) after an exercise intervention. Additional analyses showed that correcting the gait quality characteristics for walking speed did not affect our results (Figure 3 and Appendix A, Figure A1 and Figure A2).

## 5. Conclusions

Daily life gait quality characteristics have limited responsiveness to change over a year in older people who experienced a fall or participated in a balance exercise program. These findings highlight the limitations of using daily life gait quality characteristics as outcomes for studies with relatively healthy older adults over longer time frames.

## Figures and Tables

**Figure 1 sensors-23-00101-f001:**
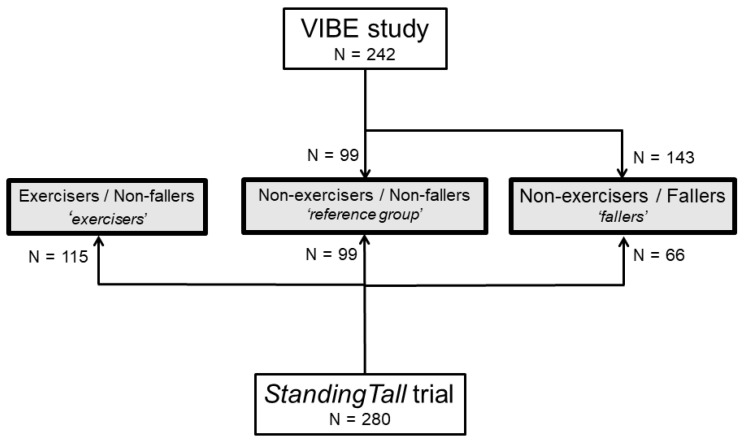
Overview of participants with available data in the VIBE cohort and *StandingTall* trial on group allocation.

**Figure 2 sensors-23-00101-f002:**
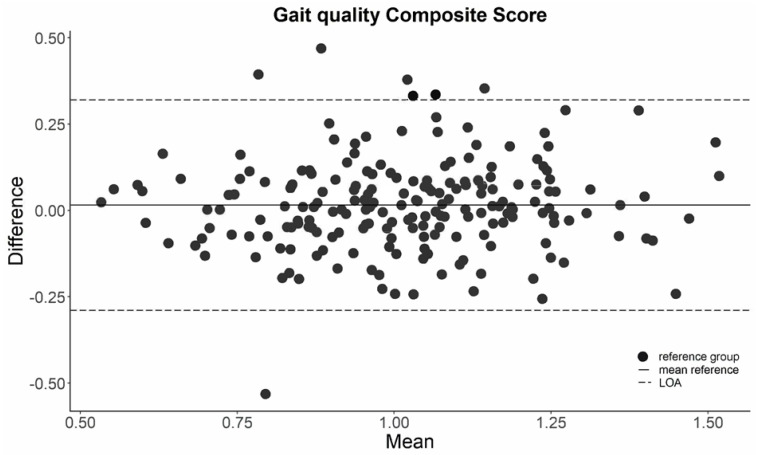
Bland–Altman plot of the gait quality composite score in the reference group. LOA = limit of agreement.

**Figure 3 sensors-23-00101-f003:**
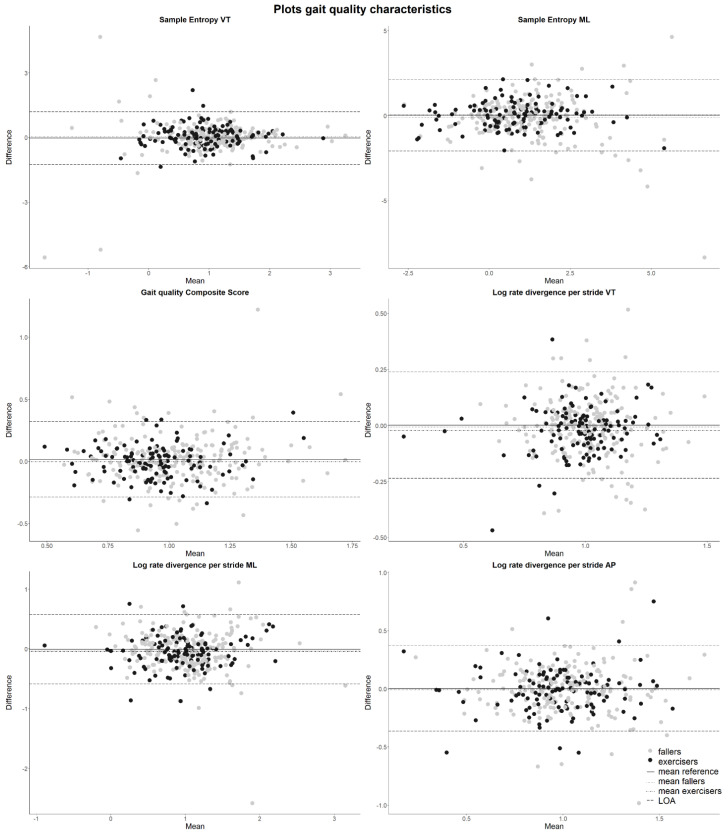
Bland–Altman plots of gait quality characteristics adjusted for walking speed with LOA. Positive differences in sample entropy in ML and gait quality composite and negative differences in sample entropy in VT or logarithmic rate of divergence in VT, ML, or AP indicate an increase in gait quality over one year. The continuous line is the mean change in the reference group. Long dashed lines are the LOA determined in the reference group. The dotted line (mean change in the fallers) and dot-dashed line (mean change in the exercisers) are sometimes overlapping the continuous line (mean change in the reference group).

**Table 1 sensors-23-00101-t001:** Baseline descriptive participant characteristics of the total sample, the reference group, fallers, and exercisers.

Variable	Total Sample	Reference Group	Fallers	Exercisers
Number of participants [N; (%)]	522	198 (37.9%)	209 (40.0%)	115 (22.0%)
Number of females [N; (%)]	352 (67.4%)	129 (65.2%)	144 (68.9%)	79 (68.7%)
Age [years; Median (IQR)]	**73.3 (69.9, 78.2)**	73.2 (69.4, 79.2)	71.9 (68.5, 76.3) **↕**	74.9 (72.3, 78.9) **↕**
Body height [cm]	166.48 (9.32)	166.82 (9.31)	167.44 (9.67)	164.13 (8.31) ↓
Body weight [kg]	73.64 (13.71)	73.52 (13.5)	73.39 (13.2)	74.28 (15.1)
Body Mass Index [kg/m^2^]	**26.55 (4.45)**	26.39 (4.3) ↔	26.19 (4.4)	27.48 (4.8) ↑
At least one fall in the past 12 months [N; (%)]	**206 (39.5)**	58 (29.3)	105 (50.2) **↕**	43 (37.4)
Fall risk *	2.36 (1.27)/0.81 (0.85)	2.30 (1.2)/0.70 (0.9)	2.41 (1.3)/1.0 (1.0)	NA/0.80 (0.8)
Knee extension torque [Nm]/knee force [kg]				
females **	152.47 (50.06)/26.20 (8.57)	148.7 (48.7)/25.98 (8.1)	154.9 (51)/25.30 (9.3)	NA/26.88 (8.6)
males **	202.60 (60.63)/42.15 (11.58)	205.26 (53.6)/40.44 (11.1)	200.57 (66.1)/41.71 (11.1)	NA/44.09 (12.3)
Cognitive function ***	31.25 (2.27)/**26.48 (2.33)**	31.14 (1.5)/26.95 (2.1)	31.32 (2.7)/26.27 (2.5) ↔	NA/26.18 (2.4) ↓
Number of participants with depressive symptoms [N] ****	96/1	29/0	65/1	NA/0
Handgrip strength [kg]				
females	51.09 (10.01)	50.78 (9.3)	51.28 (10.5)	NA
males	77.21 (13.71)	76.18 (13.6)	78.04 (13.9)	NA
Sit To Stand [s]	13.77 (40.70)	11.49 (3.8)	16.93 (64.1)	11.92 (3.7)
Walking time [min/day]	81.84 (30.11)	79.01 (28.8)	83.43 (29.9)	83.98 (32.8)
Exercise time [min/week]	NA	NA	NA	84.02 (56.74, 113.34)

Values represent means (standard deviation) unless noted otherwise. For fall risk, knee extension torque and cognitive function variables are reported for the VIBE cohort and StandingTall trial separately (VIBE/StandingTall) because of slightly different assessment methods. Bold values denote a significant main effect of group (*p*-values below 0.05). * VIBE: QuickScreen score; StandingTall: Physiological Profile Assessment score; ** VIBE: knee torque; StandingTall: knee force; *** VIBE: Mini-Mental State Examination; StandingTall: Montreal Cognitive Assessment; **** VIBE: Geriatric Depression Scale-15; StandingTall: Patient Health Questionnaire-9. **↕**: significantly different from all other groups; ↓: significantly lower than all other non-marked groups; ↑: significantly higher than other non-marked groups; ↔: not significantly different from other non-marked groups.

**Table 2 sensors-23-00101-t002:** Intraclass correlation coefficients (ICC) with 95% confidence interval, upper and lower limits of agreement (LOA_low_ and LOA_up_), and percentage fallers and exercisers that changed in gait quality beyond the limits of agreement.

	ICC [95% CI]	LOA_low_	LOA_up_	Fallers beyond LOA [%]	Exercisers beyond LOA [%]
Walking speed	0.68 [0.62, 0.74]	−0.26	0.24	3.35	1.74
Sample entropy VT	0.79 [0.74, 0.83]	−1.26	1.20	1.91	0.87
Sample entropy ML	0.77 [0.72, 0.82]	−2.08	2.13	4.78	0.87
Log rate of divergence per stride VT	0.69 [0.62, 0.75]	−0.23	0.24	2.87	2.61
Log rate of divergence per stride ML	0.85 [0.82, 0.88]	−0.59	0.57	2.39	2.61
Log rate of divergence per stride AP	0.77 [0.72, 0.81]	−0.36	0.37	1.91	2.61
Gait quality composite score	0.79 [0.74, 0.83]	−0.29	0.32	4.31	2.61

**Table 3 sensors-23-00101-t003:** Mean change in gait quality characteristics (corrected for walking speed) between baseline and follow-up in the reference group, fallers, and exercisers and their effect sizes.

	Mean Difference Reference	t	Cohen’s d	Mean Difference Fallers	t	Cohen’s d	Mean Difference exercisers	t	Cohen’s d
Walking speed	−0.01	−1.01	−0.06	−0.01	−0.87	−0.05	**−0.04 ****	−3.81	−0.22
Sample Entropy VT	−0.03	−0.72	−0.03	0.01	0.26	0.02	0.05	0.96	0.07
Sample Entropy ML	0.03	0.39	0.02	−0.11	−1.24	−0.07	0.05	0.64	0.03
Log rate divergence per stride VT	0.00	0.26	0.01	−0.01	−0.76	−0.04	**−0.02 ***	−2.19	−0.13
Log rate divergence per stride ML	−0.01	−0.30	−0.01	−0.04	−1.59	−0.07	**−0.05 ***	−2.02	−0.10
Log rate divergence per stride AP	0.00	0.33	0.02	0.00	−0.32	−0.02	−0.01	−0.64	−0.04
Gait quality Composite Score	0.02	1.62	0.08	0.02	1.24	0.07	0.00	−0.28	−0.02

Significant changes are marked in bold. * = *p* < 0.05, ** = *p* < 0.001. VT = vertical, ML = mediolateral, AP = anteroposterior.

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
