# Peer review of "Responsiveness of Daily Life Gait Quality Characteristics over One Year in Older Adults Who Experienced a Fall or Engaged in Balance Exercise"

_sensors, 2022, doi:10.3390/s23010101_

Round 1

Reviewer 1 Report

I have reviewed the manuscript and I don't have any comments on the presented calculation methods, however I suggest authors to provide more information with references regarding the relevance of their study to both clinicians and general population. 

Reviewer 2 Report

This study aimed to test whether accelerometer-based characteristics of daily life gait quality are reliable and responsive to changes over a year in older adults who have experienced falls or exercise interventions. This is an interesting topic. Here are a few issues in the manuscript that require attention/clarification/justification.

1. Statistics require more detailed analysis.

2. Does disease and lifestyle affect the experiment?

3. The content of the conclusion part is too small, and the research results of this paper need to be introduced.
